# Efficacy of a Covalent Microtubule Stabilizer in Taxane-Resistant Ovarian Cancer Models

**DOI:** 10.3390/molecules26134077

**Published:** 2021-07-03

**Authors:** Samantha S. Yee, April L. Risinger

**Affiliations:** 1Department of Pharmacology, The University of Texas Health Science Center at San Antonio, Floyd Curl Drive, San Antonio, TX 78229, USA; yees3@livemail.uthscsa.edu; 2Mays Cancer Center, 7979 Wurzbach Road, San Antonio, TX 78229, USA

**Keywords:** microtubule stabilizers, taxanes, drug resistance, natural products, ovarian cancer, metastasis, covalent drugs, anticancer agents, murine model

## Abstract

Ovarian cancer often has a poor clinical prognosis because of late detection, frequently after metastatic progression, as well as acquired resistance to taxane-based therapy. Herein, we evaluate a novel class of covalent microtubule stabilizers, the C-22,23-epoxytaccalonolides, for their efficacy against taxane-resistant ovarian cancer models in vitro and in vivo. Taccalonolide AF, which covalently binds β-tubulin through its C-22,23-epoxide moiety, demonstrates efficacy against taxane-resistant models and shows superior persistence in clonogenic assays after drug washout due to irreversible target engagement. In vivo, intraperitoneal administration of taccalonolide AF demonstrated efficacy against the taxane-resistant NCI/ADR-RES ovarian cancer model both as a flank xenograft, as well as in a disseminated orthotopic disease model representing localized metastasis. Taccalonolide-treated animals had a significant decrease in micrometastasis of NCI/ADR-RES cells to the spleen, as detected by quantitative RT-PCR, without any evidence of systemic toxicity. Together, these findings demonstrate that taccalonolide AF retains efficacy in taxane-resistant ovarian cancer models in vitro and in vivo and that its irreversible mechanism of microtubule stabilization has the unique potential for intraperitoneal treatment of locally disseminated taxane-resistant disease, which represents a significant unmet clinical need in the treatment of ovarian cancer patients.

## 1. Introduction

Annually, approximately 22,000 women are diagnosed with ovarian cancer in the United States of which 14,000 will succumb to the disease [1]. Ovarian cancer is a heterogeneous disease with few targeted therapies and a high rate of metastasis [2]. It is the most malignant disease out of the gynecological cancers. Patients are often diagnosed at late stages after dissemination to other organ sites, such as the omentum, liver, and spleen have occurred [3,4,5]. Unfortunately, due to current methods of detection, about 60% of patients are diagnosed with advanced ovarian cancer [6]. The standard of care for ovarian cancer patients consists of combination chemotherapy of paclitaxel, a plant-derived natural product, and a platinum-based agent administered intravenously or intraperitoneally [7,8,9]. Although this treatment is effective initially, about 80% of patients with advanced ovarian cancer relapse after combination chemotherapy, with a median progression free survival of approximately 12–18 months and 5-year survival rate of 48% [1]. One reason for this relapse is that up to 70% of patients acquire taxane resistance within this timeframe, often due to expression of multidrug efflux transporters, including P-glycoprotein (Pgp) [10,11,12]. The development of a microtubule stabilizer that retains efficacy in taxane-resistant ovarian cancers would be significant as microtubule disruption is a proven target in this disease [9,13]. Further, several therapeutics derived from natural products have demonstrated efficacy in ovarian cancer models [14,15]. 

The epoxytaccalonolides are a novel class of plant-derived microtubule stabilizers that bind covalently and irreversibly to tubulin through a specific interaction between their C-22,23 epoxide with Asp226 on β-tubulin [16,17]. Although the epoxytaccalonolide AF (Figure 1A) has demonstrated antitumor efficacy in flank xenograft breast cancer models with systemic administration, it has a narrow therapeutic index and short serum half-life of 45 min [18,19]. Other epoxytaccalonolides, including AJ (Figure 1B), have no therapeutic index with systemic injection, but do demonstrate potent and highly persistent antitumor efficacy when directly injected into the local tumor microenvironment, even in Pgp-expressing tumors that are resistant to the taxanes [20]. These findings suggest that covalent microtubule stabilizers, including the taccalonolides, may be optimal for the treatment of locally metastatic, taxane-resistant ovarian cancer through intraperitoneal (i.p.) administration, which is a clinically validated method of treatment for this disease [21]. When chemotherapeutic drugs are administered by i.p. injection it decreases systemic exposure and associated toxicity, and also has a therapeutic advantage in locally metastatic ovarian cancer, providing a high peritoneal to plasma ratio [22,23,24]. Further, some clinical trials have demonstrated a survival advantage in ovarian cancer with administration of chemotherapy by i.p. injection as compared to intravenous-based treatment [25,26,27,28]. We hypothesized that the taccalonolides may have optimal efficacy in locally metastatic ovarian cancer models where i.p. administration would allow for targeted delivery of this irreversible microtubule stabilizer.

Herein, we test the hypothesis that taccalonolide AF effectively circumvents clinically relevant models of taxane resistance in ovarian cancer models both in vitro and in vivo due to its irreversible target engagement leading to a high degree of cellular persistence and potent antiproliferative and cytotoxic effects. We demonstrate that taccalonolide AF has superior antitumor efficacy as compared to paclitaxel in a Pgp-expressing NCI/ADR-RES drug-resistant ovarian cancer flank xenograft model. In addition, we show that AF retains efficacy in a disseminated metastatic model of this drug-resistant ovarian cancer cell line with a broader therapeutic window than against a breast cancer flank xenograft model [18,29]. 

## 2. Results

### 2.1. Taccalonolide AF Retains Efficacy in Taxane-Resistant Ovarian Cancer Models In Vitro

Multidrug resistance (MDR) is a prominent mechanism of resistance to clinically approved therapeutics in ovarian cancer patients [10,30,31]. A major mechanism of MDR is enhanced efflux of drugs, including increased expression of *MDR1*, which encodes the drug efflux transporter, P-glycoprotein (Pgp). Prior studies have demonstrated that paclitaxel is a Pgp substrate and increased *MDR1* expression leads to clinical resistance to taxanes [32]. To determine whether the covalent microtubule stabilizer, taccalonolide AF (Figure 1A), retains efficacy in Pgp-expressing, taxane-resistant ovarian cancer models in vitro, we selected two isogenic cell line pairs that have been reported to express high levels of *MDR1*. The SK-OV-3-MDR-1-6/6 line is derived from transduction of the parental SK-OV-3 ovarian cancer line with *MDR1* [33]. The NCI/ADR-RES cell line was generated by selection for resistance to adriamycin and later found to be a *MDR1*-expressing derivative of the OVCAR-8 line [34]. Indeed, we found that the SK-OV-3-MDR-1-6/6 and NCI/ADR-RES cell lines, respectively, have 1600 and 9000-fold increased *MDR1* expression than their parental cell lines (Figure 2A,D). This level of *MDR1* expression is somewhat higher than those observed in a cohort of drug-resistant ovarian cancer patients, which were found to be up to 653-fold higher than the SK-OV-3 cell line [35], suggesting that our choice of cell lines provides stringent models for the evaluation of *MDR1*-mediated drug resistance in ovarian cancer.

Once we identified the SK-OV-3-MDR-1-6/6 and NCI/ADR-RES cell lines as *MDR1-*expressing ovarian cancer models, we assessed the potency and efficacy of taccalonolide AF in these lines and their respective isogenic parental lines using the sulforhodamine B assay. We determined the concentration of AF or paclitaxel that inhibited growth by 50% (GI_50_) in each cell line and calculated relative resistance values by dividing the potency in each *MDR1*-expressing line by its potency in the respective parental line (Table 1). While NCI/ADR-RES cells were over 870-fold resistant to paclitaxel as compared to the parental OVCAR-8 line, they were only 124-fold resistant to AF (Table 1) suggesting AF is less susceptible to Pgp-mediated resistance as compared to paclitaxel. These results were consistent in the other cell line pair where the SK-OV-3-MDR-1-6/6 line was 90-fold resistant to paclitaxel but only 11-fold resistant to AF (Table 1). Similar results were obtained when calculating the concentration of each compound that caused total growth inhibition (TGI) over the respective treatment period (Table 2). The NCI/ADR-RES cells were 458-fold resistant to paclitaxel and only 30-fold resistant to AF by this measure, whereas SK-OV-3-MDR-1-6/6 cells were 98-fold resistant to paclitaxel and 15-fold resistant to AF (Table 2). These data demonstrate that while taccalonolide AF is somewhat susceptible to *MDR1*-mediated drug resistance when the drug efflux transporter is expressed at levels thousands of times greater than parental lines, that this degree of drug resistance is 7–8-fold less than for paclitaxel.

The greater degree of resistance observed for both AF and paclitaxel in the NCI/ADR-RES cells as compared to the SK-OV-3-MDR-1-6/6 cells correlates with the relative levels of *MDR1* expression in these two lines (Table 1, Figure 2). To test the hypothesis that Pgp-mediated drug efflux is the primary mechanism of paclitaxel resistance in both of these resistant cell lines, we utilized verapamil as a first-generation Pgp inhibitor [36]. Concentrations of 5–10 µM verapamil were identified as maximal concentrations that did not inhibit growth of NCI/ADR-RES or SK-OV-3-MDR-1-6/6 cells on their own. Indeed, we found that verapamil co-treatment shifted the concentration-response curves of paclitaxel and AF in SK-OV-3-MDR-1-6/6 cells to overlap with the parental SK-OV-3 cell line (Figure 2B,C) such that the GI_50_ and TGI values for the drug-resistant line with verapamil were in line with the values in the parental line (Table 1 and Table 2). A similar trend was observed with the NCI/ADR-RES cell line (Figure 2E,F). However, this line retained some relatively minor resistance to both paclitaxel (17-fold) and taccalonolide AF (2.6-fold), demonstrating that even high concentrations of verapamil were not sufficient to completely reverse resistance in this high *MDR1*-expressing line. These data were supported by genetic knockdown experiments where *MDR1* siRNA restored the sensitivity of these Pgp-expressing lines to paclitaxel in particular (data not shown). Together, these data support previous reports that the primary mechanism of resistance in both the NCI/ADR-RES and SK-OV-3-MDR-1-6/6 cell lines is due to Pgp-mediated drug efflux and that AF is less susceptible than paclitaxel to this resistance mechanism. We hypothesize that taccalonolide AF is less susceptible to Pgp-mediated drug resistance than other microtubule targeting agents due to its irreversible target engagement and that this may provide a unique potential in the treatment of *MDR1-*expressing, taxane-resistant ovarian cancers.

It is well-documented that ovarian cancer is a heterogenous disease and that the high-grade serous subtype of ovarian cancer (HGSOC) is the predominant form of the disease observed in the clinic [37]. While the OVCAR-8 cell line used above is characterized as HGSOC, we wanted to further evaluate the potency and efficacy of taccalonolide AF in a broader panel of molecularly diverse HGSOC cell lines [38]. Taccalonolide AF effectively inhibited the proliferation of each of the HGSOC lines at concentrations ranging from 6–41 nM (Table 3). The relative sensitivity of these cell lines to AF was, for the most part, consistent with paclitaxel and was not correlated with *MDR1* expression, which was low in each of these lines. Of note, AF was significantly less potent than paclitaxel in OV-90 and OVCAR-3 cells, which have a similar expression of tumor associated antigens [39]. Together, these results indicate that AF has activity against a diverse subset of clinically relevant human ovarian cancer models with the greatest advantage over paclitaxel in *MDR1-*expressing models.

### 2.2. Taccalonolide AF Has Potent and Persistent Effects in Ovarian Cancer Cell Lines Following Acute Exposure

While taccalonolide AF has advantages over paclitaxel in Pgp-expressing ovarian cancer cells, it has potency comparable to paclitaxel in most ovarian cancer models when the cells are continuously exposed to the drug (Table 1, Table 2 and Table 3). However, we hypothesized that the irreversible binding of the epoxytaccalonolides to tubulin would provide an additional advantage over paclitaxel when cells were only subjected to an acute drug exposure prior to drug washout, which is more relevant to in vivo conditions where drugs are cleared over time. To compare the cellular persistence of taccalonolide AF to paclitaxel in vitro, we first exposed SK-OV-3 cells, which were equally sensitive to these drugs in the 48 h SRB assay (Table 1), to 50 nM of each drug for just 1–3 h followed by drug washout and allowed colonies to form over a period of an additional 12 days in fresh medium lacking the drug (Figure 3A). These concentrations were chosen because they caused an average of 30–40% cytotoxicity over the full 48 h period of the SRB assay, although there was no evidence of toxicity during the acute 1–3 h drug treatment period.

A two-way ANOVA with a Dunnett’s multiple comparisons test was conducted where (drug * time) interaction (F (6, 21) = 3.494, *p* = 0.0148), time (F(3, 21) = 5.132, *p* = 0.0081), and drug alone (F(2, 21) = 14.15, *p* = 0.0001) were observed to have statistical significance (Figure 3B). When drugs were only present for 1–2 h prior to washout, no statistical difference was observed in number of colonies between control and PTX treated groups (*p* = 0.1888 and *p* = 0.7061) or between control and AF treated groups (*p* = 0.7860 and *p* = 0.2239) (Figure 3B and Appendix A). However, when the duration of drug treatment was increased to 2.5–3 h, AF caused a statistically significant reduction in number of colonies as compared to control (*p* < 0.0001 and *p* = 0.0098). In contrast, there was no statistical difference between PTX treated and control conditions (*p* = 0.6986 and *p* = 0.1400) (Figure 3B and Appendix A). Together, these data demonstrate that taccalonolide AF has advantages over paclitaxel even in ovarian cancer cells that are equivalently sensitive to both paclitaxel and AF in traditional antiproliferative assays. The long-term, highly persistent inhibition of colony formation by AF over a 12 day period after drug washout after only 2–3 h of acute drug treatment are likely a result of their irreversible target engagement. These findings are important as previous studies have associated in vitro cellular persistence with in vivo efficacy and even used this criteria in the selection of an optimal clinical candidate in the case of the microtubule destabilizer, eribulin [51].

### 2.3. Taccalonolide AF Has Antitumor Efficacy against a Taxane-Resistant Ovarian Cancer Flank Xenograft Model

Based on our in vitro studies, we further evaluated the in vivo antitumor efficacy of taccalonolide AF against the NCI/ADR-RES taxane-resistant ovarian cancer flank xenograft model. Once tumors were palpable, animals were dosed with 20 mg/kg paclitaxel on days 0 and 4 (a dose that has demonstrated antitumor efficacy in a drug sensitive model [18]) or 2 mg/kg taccalonolide AF on days 0, 4, and 7 (a maximal tolerated dose (MTD) [18]), by i.p. injection. We found that a total dose of 6 mg/kg taccalonolide AF slowed the growth of tumors as compared to animals treated with paclitaxel or untreated controls (Figure 4). A two-way ANOVA with treatment as between-subjects factor (levels: control, paclitaxel, taccalonolide AF) and days as within-subjects factor (levels: 0–20) showed a statistically significant treatment * days interaction (F(12, 132) = 2.16, *p* = 0.017). Thus, tumor growth did not show the same progression over time in the three treatment conditions. Comparing each of the treatments with control and with each other using two-way ANOVAs showed a statistically significant treatment * day interaction for taccalonolide AF vs. control (F (6, 96) = 2.68, *p* = 0.019) and for taccalonolide AF vs. paclitaxel (F (6, 84) = 5.67, *p* < 0.0001), but not for paclitaxel vs. control (F (6, 84) = 0.1913, *p* = 0.98). Together, these data demonstrate that tumor growth progressed less rapidly with taccalonolide AF than with the other two treatments (Figure 4). Further, we observed no gross signs of toxicity, i.e., weight loss, in taccalonolide AF-treated animals indicating that this effective dose of taccalonolide AF was well tolerated in this model.

### 2.4. Taccalonolide AF Inhibits Micrometastasis from an i.p. Disseminated Ovarian Cancer Model

The majority of women are diagnosed with ovarian cancer at advanced stages where the disease is locally disseminated within the peritoneal cavity. This clinical presentation has led to the administration of chemotherapy by localized i.p. injection where prolonged survival has been observed, supporting this route of administration as a promising strategy for advanced ovarian cancer patients [52]. We therefore evaluated the efficacy of taccalonolide AF against an i.p. disseminated NCI/ADR-RES model based on a prior study demonstrating i.p. injection of this line promoted metastasis throughout the peritoneal cavity [53]. However, as opposed to gross metastasis measured by luciferin-based in vivo imaging we focused on quantifying the extent of micrometastasis to organs within the peritoneal cavity, including the spleen, liver, heart, ovaries with uterus, and lung using a quantitative PCR-based approach [54]. This approach, which utilizes a standard curve of cancer cells spiked into naive organ tissue (Appendix A), allows for a rigorous analysis of the extent of metastasis to different organ sites and the ability of compounds to inhibit the formation of these lesions. 

We first determined the timing of micrometastasis of i.p. injected NCI/ADR-RES cells to organs of the peritoneal cavity by harvesting organs prior to (day 0), or 11 to 40 days after cell injection. As expected, on day 0 no metastatic cells were found in the mouse organs as determined by the lack of detection of human GAPDH in samples where mouse GAPDH values served as normalization controls (Appendix A). In contrast, NCI/ADR-RES cells were detected in many of the organs harvested on days 11 (Figure 5A) and 40 (Figure 5B) after cell injection. The number of cells present in each sample was extrapolated from a comparison of the Ct values for human GAPDH in each organ as compared to the standard curve generated when known numbers of these cells were spiked into naïve tissue (Appendix A). This method allowed for the detection of NCI/ADR-RES cells in all tissues other than the heart 11 days after cell injection (Figure 5A and Appendix A). By day 40, the number of NCI/ADR-RES cells in all tissues increased further (Figure 5B). Overall, we found that the spleen was the organ with the greatest and most consistent extent of metastasis (Figure 5B), an organ site that is relevant in the human disease [55]. 

To evaluate the effects of microtubule stabilizers on these micrometastatic lesions, we treated mice with 1.5 mg/kg taccalonolide AF or 20 mg/kg paclitaxel by i.p. injection 11 days after NCI/ADR-RES cells were injected into the peritoneal cavity with a subsequent dose of each drug administered 3 days later. We slightly lowered the dose of taccalonolide AF in this study as compared to our flank xenograft model to test the hypothesis that doses lower than the MTD would have efficacy with localized injection. No weight loss or other signs of toxicity were observed over the duration of the trial (Appendix A). Animals were sacrificed and organs were harvested 40 days after cell injection and the extent of micrometastasis in each organ quantified and compared to control animals. We focused on the quantification of NCI/ADR-RES cells in the spleen under the different treatment conditions as this was the most consistent site of local metastasis for this model. A one-way ANOVA with Dunnett’s multiple comparisons test comparing the treatment and control conditions showed a significant treatment effect (F(2, 12) = 4.150, *p* = 0.0427) with a statistically significant effect of taccalonolide AF vs. control (*p* = 0.0254) but not for paclitaxel vs. control (*p* = 0.2479) (Figure 5C and Appendix A). We also conducted a non-parametric test (Kruskal–Wallis followed by Dunn’s multiple comparisons test), which unlike an ANOVA does not assume normality and homogeneity of variance and obtained results similar to those of the ANOVA analyses. Further, we conducted a one-way ANOVA comparing the number of NCI/ADR-RES cells in the spleen on day 11, when the first dose of AF was administered, as compared to control or AF-treated animals on day 40. A Tukey’s multiple comparisons analysis showed a significant treatment effect (F(2, 10) = 4.630, *p* = 0.0377) with a statistically significant effect of taccalonolide AF vs. control (*p* = 0.0323) but not for either AF or control on day 40 vs. day 11 (*p* = 0.2448, 6626) (Figure 5D). Therefore, while taccalonolide AF treatment significantly reduced the number of NCI/ADR-RES cells in the spleen on day 40 as compared to vehicle treated animals, we could not definitively say whether this was due to a decrease in the number of cells as compared to the time of drug administration or simply an inhibition of proliferation for cells already present in the spleen on day 11 (Figure 5D). It is important to note that although we did not detect a statistical difference between NCI/ADR-RES cells in the spleen between the time AF was first administered and the end of the trial, we did observe two animals with no detectable metastasis in the spleen after taccalonolide AF treatment (Figure 5D). Indeed, in one AF-treated animal, no NCI/ADR-RES were detected in any of the five organs harvested. We did not detect statistically significant treatment differences for the number of NCI/ADR-RES cells in other organs, likely due to the high degree of variability of cell number within organ sites other than the spleen even under control conditions (Appendix A). Together, these data demonstrate that taccalonolide AF is more effective than paclitaxel at inhibiting localized metastasis, particularly to the spleen, which is a clinically relevant site of ovarian cancer metastasis and the organ with the highest degree of metastasis in this model. 

## 3. Discussion

The taxane microtubule stabilizers are a mainstay in the treatment of adult solid tumors, including ovarian cancer where paclitaxel in combination with a platinum-based agent is first line therapy. Although highly effective initially, up to 70% of patients acquire taxane resistance within 18 months [10,11,12,56,57]. Therefore, it is essential to identify compounds that can overcome taxane resistance for the effective treatment of these women. The C-22,23-epoxytaccalonolides, including taccalonolide AF, are microtubule stabilizers that covalently and irreversibly bind to microtubules, providing an advantage in clinically relevant models of drug resistance [20,58]. Herein, we show that taccalonolide AF is less susceptible than paclitaxel to resistance mediated by increased expression of the *MDR1*-encoded P-glycoprotein drug efflux pump, a major form of taxane resistance in the clinic [10,30,31]. Additionally, the irreversible target engagement of AF leads to superior cellular persistence after short periods of drug exposure and washout as compared to paclitaxel, suggesting an additional advantage in vivo where the drugs are subject to clearance. Indeed, these attributes are underscored by the potency and efficacy of AF in a taxane-resistant ovarian flank xenograft model.

The clinical development of covalent microtubule stabilizers for cancer treatment will likely involve tumor targeting strategies to decrease systemic toxicity and improve the therapeutic index. One effective tumor targeting strategy involves antibody-drug conjugates (ADCs) that commonly utilize potent microtubule destabilizers as cytotoxic payloads [59]. While we propose that the irreversible microtubule engagement of the taccalonolides could provide a novel ADC payload that is worth pursuing, a more immediate targeting strategy is to take advantage of i.p. drug delivery for the localized treatment of metastatic ovarian cancer. With the discovery that the most frequent type of ovarian cancer, high-grade serous ovarian carcinoma (HGSOC), actually originates in the fallopian tube, it is clear that this deadly disease has effectively undergone localized metastasis even at the time of diagnosis [60]. 

The efficacy of microtubule stabilizers as first line therapy in HGSOC combined with the rapid development of drug resistance and the ability to target locally disseminated disease with i.p. administration makes HGSOC an attractive candidate for the use of the taccalonolides. We evaluated the efficacy of AF against localized metastases in a taxane-resistant ovarian cancer model using a highly quantitative methodology that allowed both a characterization of the temporal spread of this model to discrete organs within the peritoneal cavity, as well as a rigorous evaluation of drug efficacy. Collectively, we demonstrate that the NCI/ADR-RES model consistently establishes within the spleen when disseminated within the peritoneal cavity and that AF, but not paclitaxel, significantly reduces this growth, essentially eliminating detectable metastases in some animals. Together, these findings suggest that covalent microtubule stabilizers could be an important tool in the treatment of taxane-resistant ovarian cancers, particularly with localized i.p. injection to target disseminated disease.

## 4. Materials and Methods

### 4.1. Chemicals Compounds

Paclitaxel was obtained from Sigma-Aldrich (T1912-25MG, St. Louis, MO, USA) and brought up in 200 proof EtOH. Taccalonolide AF was semi-synthesized as previously described and brought up in 200 proof EtOH [16]. (±)-Verapamil hydrochloride was obtained from Sigma (V4629, St. Louis, MO, USA) and brought up in water. 

### 4.2. Human Ovarian Cancer Cell Lines

OV-90 (CRL-11732) and ES-2 (CRL-1978) cell lines were obtained from Dr. Gangadhara Sareddy (UT Health San Antonio) while OVCAR-5, OVCAR-3, OVCAR-8, and NCI/ADR-RES cells were obtained directly from the National Cancer Institute’s Developmental Therapeutics, program (NCI DTP). OVSAHO, Kuramochi, and JHOS-4 high grade serous ovarian cancer cell lines were a gift from Dr. Ronny Drapkin (UPenn). SK-OV-3 (HTB-77) ovarian cancer cells were obtained from ATCC and validated by STR profiling (Genetica). SK-OV-3 cells stably overexpressing Pgp by adenoviral-mediated expression of *MDR1* were obtained from Dr. Susan Kane and subcloned by limiting dilution to isolate the single-cell clones utilized in these studies as SK-OV-3-MDR-1-6/6 [33]. SK-OV-3 and SK-OV-3-MDR-1-6/6 cells were grown in BME media with Earle’s salts (Gibco, Grand Island, NY, USA) with 10% FBS (Corning, Corning, NY, USA), 1 × final 1% GlutaMax™ Supplement (Gibco, Grand Island, NY, USA), and 50 µg/mL gentamicin (Gibco, Grand Island, NY, USA). ES-2, OVSAHO and Kuramochi cells were cultured in RPMI 1640 media (Corning, Corning, NY, USA) with 10% FBS and 50 µg/mL gentamicin. OVCAR-3 cells were grown in RPMI 1640 media with 20% FBS and 50 µg/mL gentamicin. OV-90 cells were grown in a 1:1 mixture of MCDB 105 medium containing a final concentration of 1.5 g/L sodium bicarbonate (MilliporeSigma, St. Louis, MO, USA) and Medium 199 containing a final concentration of 2.2 g/L sodium bicarbonate (Thermo Fisher Scientific, Waltham, MA, USA) with 15% FBS. OVCAR-5 and JHOS-4 cells were cultured in DMEM, high glucose (Gibco, Grand Island, NY, USA) with 10% FBS and 50 µg/mL gentamicin. All cell lines were grown at 37 °C in an incubator with 5% CO_2_. Cells were tested regularly for mycoplasma contamination. 

### 4.3. Antiproliferative and Cytotoxicity Assay

The sulforhodamine B (SRB) assay was utilized to examine the antiproliferative and cytotoxic effects of the compounds. Cells were seeded in 96-well plates (Corning, Corning, NY, USA) and treated in triplicate with each concentration of compound or vehicle control for 48 h in a final volume of 200 µL. Cells were fixed with 10% trichloroacetic acid (Sigma-Aldrich, St. Louis, MO, USA), protein stained with SRB dye (Sigma-Aldrich, St. Louis, MO, USA) and total protein per well quantified by absorbance at 560 nm after solubilizing dye in 10 mM Tris. Concentration-response curves were generated by non-linear regression analysis using Prism version 8 (GraphPad software, San Diego, CA, USA) and the GI_50_ of each compound was calculated as the concentration that caused a 50% growth inhibition in comparison to vehicle control from a minimum of 3 independent experiments (n = 3). The TGI was defined as the concentration that completely inhibited growth of cells over the course of treatment. SK-OV-3-MDR-1-M6/6 and NCI/ADR-RES human taxane-resistant ovarian cancer cell lines were treated with verapamil to identify the highest concentration that would not affect growth of the cells on its own (5 µM for NCI/ADR-RES and 10 µM for SK-OV-3-MDR-1-6/6). These verapamil concentrations were used in combination with paclitaxel and taccalonolide AF to determine the contribution of Pgp-mediated drug efflux to drug potency in *MDR1-*expressing cell lines.

### 4.4. qRT-PCR

RNA was isolated from human ovarian cancer cells by Trizol and chloroform extraction. The RNA pellet was resuspended in nuclease-free water and quantified using a Nanodrop 2000. RNA was converted to cDNA with iScript Reverse Transcription Supermix for RT-qPCR (Bio-Rad, Hercules, CA, USA) and qRT–PCR performed using iTaq Universal SYBr Green Supermix (Bio-Rad, Hercules, CA, USA). Human Pgp primers (Sigma Aldrich, St. Louis, MO, USA) were generated based on the literature [61] and specificity confirmed using NCBI Primer-BLAST. Human GAPDH primers: 5′-GCAAATTCCATGGCACCGT-3′ and 5′-TCGCCCCACTTGATTTTGG-3′. Further, RNA was isolated from murine spleen, liver, lung, heart, and ovaries with uterus with or without human ovarian cancer cells by Trizol in a total volume of 3 mL and chloroform extraction. Organs were minced using a razor blade and 500 µL volumes were utilized for each sample for RNA extraction. Human-specific GAPDH primers for the in vivo trial were obtained from [54]. Mouse GAPDH primer sequences utilized for normalization: 5′-CGACTTCAACAGCAACTCCCACTCTTCC-3′ 5′-TGGGTGGTCCAGGGTTTCTTACTCCTT-3′. Samples were run as technical duplicates and were only accepted if replicates were within 2 Ct of one another as a measure of rigor and reproducibility.

### 4.5. Persistence Assay

SK-OV-3 human ovarian cancer cells were plated in 60 mm cell culture dishes (Corning, Corning, NY, USA). Cells were treated with 50 nM paclitaxel or taccalonolide AF, concentrations that caused 30–40% cytotoxicity in the 48 h SRB assay with no evidence of toxicity during the acute 1–3 h treatment. After this 1–3 h drug incubation, cells were washed to remove excess drug and replaced with fresh medium until visible colonies had formed 12 days later. Colonies were fixed with 20% MeOH and 0.5% crystal violet for 20–30 min. Images were captured by camera and imported into ImageJ where colonies were counted and recorded. Three independent experiments (n = 3) were conducted for each condition. 

### 4.6. Animal Care and Welfare

All the studies using mice were conducted in accordance with an approved IACUC protocol 20170208AR and in compliance with the NIH Guide for the Use of Laboratory Animals. Female athymic nude mice were purchased from Envigo and housed in an Association for Assessment and Accreditation of Laboratory Animal Care-approved facility in temperature-controlled rooms and provided food and water ad libitum. 

### 4.7. NCI/ADR-RES Flank Xenograft Model

NCI/ADR-RES human ovarian cancer cells were injected bilaterally into the flanks of female athymic nude mice with Matrigel (Corning^®^ Matrigel^®^ 356237, Corning, NY, USA). When a minimum tumor size of 100 mm^3^ was reached, indicated as day 0, mice were pair-matched and enrolled into one of three treatment groups: control (n = 9), taccalonolide AF (n = 9), or paclitaxel (n = 7). Animals were dosed on days 0 and 4 with 20 mg/kg PTX or with 2 mg/kg AF on days 0, 4, and 7 and tumor volumes measured by calipers as mm^3^ (length × width × height). Dosing was based on previous studies demonstrating the effective doses of these drugs in a taxane-sensitive model, which was also a maximally tolerated dose of AF [18]. The trial was terminated on Day 20 due to the size of control and PTX-treated tumors. Body weight and health of the mice were monitored throughout the entire trial. 

### 4.8. Disseminated Metastasis In Vivo Taxane-Resistant Ovarian Cancer Model

Approximately 8 million NCI/ADR-RES ovarian cancer cells were injected intraperitoneally in a total volume of 200 µL PBS. The spleen, left bottom liver lobe, lung, heart, and ovaries with uterus were harvested from animals 11 or 40 days after cell injection and RNA extracted for qRT-PCR analysis of human GAPDH (hGAPDH) to quantify human cells in each organ site. The processing of organs for qRT-PCR analysis and generation of the standard curve of metastatic cells per organ were performed as previously described [54]. Standard curves were generated by the quantification of hGAPDH from samples containing a known number (100, 1000, 10,000, 100,000, or 1,000,000) of NCI/ADR-RES cells spiked into each organ (spleen, left bottom liver lobe, lung, heart, and ovaries attached to the uterus) isolated from nontumored, untreated female nude mice for a total of 2–3 independent experiments from individual mice. Murine GAPDH (mGAPDH) was utilized as an internal normalization control to ensure rigor and reproducibility amongst all samples. Samples were run as technical duplicates and values were only accepted if replicates were within 2 Ct of one another. The hGAPDH mRNA transcript levels were evaluated to quantify resident human ovarian cancer cells in each organ as compared to standard curves. Data are presented from the organs of three independent animals for each condition and time point. To test the efficacy of AF and paclitaxel in this model, animals were dosed 11 days after cell injection with 20 mg/kg paclitaxel (12.5% EtOH: 12.5% Cremaphor in PBS), 1.5 mg/kg taccalonolide AF (37.5% EtOH in PBS) or vehicle (12.5% EtOH in PBS) in a total volume of 200 µL. The dosing of taccalonolide AF was reduced slightly in this experiment as compared to the flank xenograft model to test the hypothesis that localized injections could provide efficacy at doses lower than the MTD. The mice received a second dose 4 days after the initial dose for a total of 2 doses. Mice were weighed bi-weekly to monitor health and drug toxicity. Five mice were included in each drug treatment group for a total of 15 mice. Animals were euthanized 40 days after cell injection, organs (spleen, left bottom liver lobe, lung, heart, and ovaries attached to the uterus) harvested, and the number of NCI/ADR-RES cells in each organ quantified by qRT-PCR using a standard curve as described above.

### 4.9. Statistics

Prism version 8 (GraphPad software, San Diego, CA, USA) was utilized for statistics. Data is presented as mean ± SEM unless otherwise noted. Figure legends denote respective statistical analysis and post hoc test for each experiment. 

## 5. Conclusions

In summary, we demonstrate that the covalent microtubule stabilizer, taccalonolide AF, is less susceptible to Pgp-mediated resistance than paclitaxel and has a higher degree of cellular persistence after drug washout, providing an advantage in taxane-resistant ovarian cancer models both in vitro and in vivo. Further, we demonstrate an advantage of taccalonolide AF over paclitaxel in a disseminated metastatic ovarian cancer model that suggests promise for their use in the localized treatment of this deadly gynecological disease. 

## 6. Patents

S.S.Y. and A.L.R. are listed as inventors on patents on the taccalonolides that have been issued to the UT System.

## Figures and Tables

**Figure 1 molecules-26-04077-f001:**
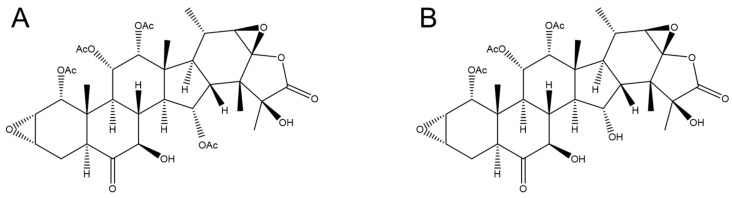
Chemical structures of the C-22,23-epoxytaccalonolides, AF (**A**) and AJ (**B**).

**Figure 2 molecules-26-04077-f002:**
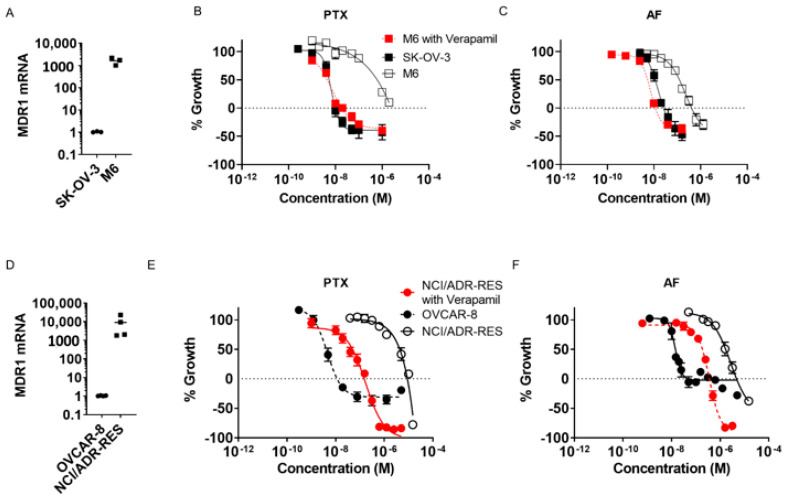
Taccalonolide AF is less susceptible to Pgp-mediated resistance than paclitaxel. (**A**) *MDR1* transcript levels in SK-OV-3 and SK-OV-3-MDR-1-6/6 (M6) cells and (**B**,**C**) concentration-response curves of paclitaxel (PTX) or taccalonolide AF with or without 10 µM verapamil. (**D**) *MDR1* transcript levels in OVCAR-8 and NCI/ADR-RES cells and (**E**,**F**) concentration-response curves of PTX or taccalonolide AF with or without 5 µM verapamil.

**Figure 3 molecules-26-04077-f003:**
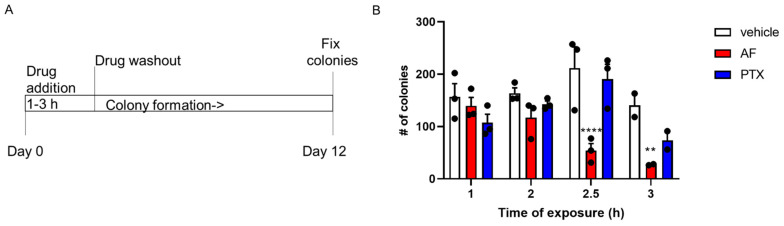
Taccalonolide AF has potent and persistent effects in ovarian cancer cells following acute exposure. (**A**) Persistence assay timeline. (**B**) Colony formation of SK-OV-3 cells after treatment with 50 nM paclitaxel (PTX) or taccalonolide AF compared to vehicle for 1–3 h (n = 2–3 independent experiments) before cells were washed and replaced with fresh media lacking drug. Closed circles represent individual data points. Statistical significance determined by two-way ANOVA with Dunnett’s post hoc test for multiple comparisons with significance compared to vehicle shown. **** *p* < 0.0001 and ** *p* = 0.0098.

**Figure 4 molecules-26-04077-f004:**
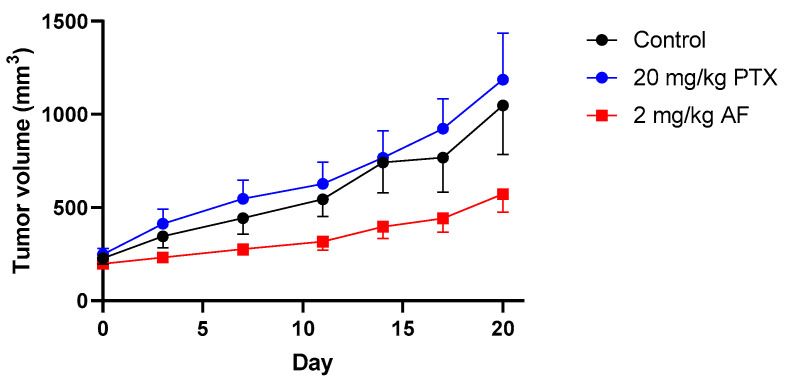
Antitumor efficacy of taccalonolide AF against the NCI/ADR-RES flank xenograft model. Animals were dosed on days 0 and 4 with 20 mg/kg PTX, or 2 mg/kg AF on days 0, 4, and 7 via i.p. injection as compared to untreated control tumors. Control (n = 9), PTX (n = 7), taccalonolide AF (n = 9). Comparison of each of the treatments with control and with each other by two-way ANOVA showed a statistically significant treatment * day interaction for taccalonolide AF vs. control (F (6, 96) = 2.68, *p* = 0.019) and for taccalonolide AF vs. paclitaxel (F (6, 84) = 5.67, *p* < 0.0001), but not for paclitaxel vs. control (F (6, 84) = 0.1913, *p* = 0.98).

**Figure 5 molecules-26-04077-f005:**
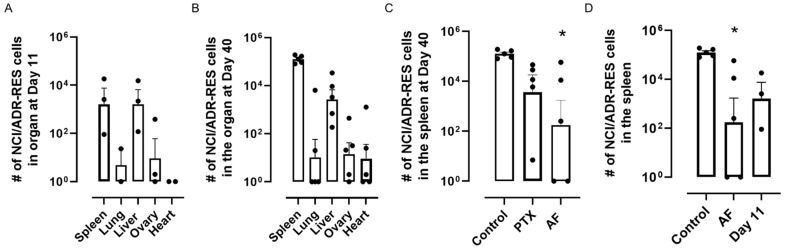
Taccalonolide AF reduces spleen micrometastases in a disseminated ovarian cancer xenograft model. (**A**) Quantification of NCI/ADR-RES cells in organs of animals 11 days after cell injections (n = 2–3 mice). (**B**) Quantification of NCI/ADR-RES cells in organs of animals 40 days after cell injections (n = 5 mice). (**C**) Quantification of NCI/ADR-RES cells in the spleens of animals treated with 20 mg/kg PTX, 1.5 mg/kg AF or vehicle on day 11 and harvested on day 40 (n = 5 mice). (**D**) Quantification of NCI/ADR-RES cells in the spleens of animals at the time of AF treatment (day 11) or with AF or vehicle at day 40. Closed circles represent individual data points. Statistical significance determined by one-way ANOVA with Dunnett’s (5C) or Tukey’s (5D) post hoc test for multiple comparisons and significant differences as compared to vehicle control are shown. * *p* < 0.05.

**Table 1 molecules-26-04077-t001:** GI_50_ (nM) of paclitaxel and taccalonolide AF in *MDR*1-expressing ovarian cancer cell lines as compared to isogenic parental lines or in the presence of the Pgp-inhibitor verapamil. Mean ± SEM n = 3–5. Relative resistance calculated as the GI_50_ of the *MDR1-*expressing cell line divided by the GI_50_ value in the isogenic parental line.

Cell Line	Paclitaxel (nM)	Relative Resistance	Taccalonolide AF (nM)	Relative Resistance
OVCAR-8	4.2 ± 0.8	-	14 ± 1	-
NCI/ADR-RES	3655 ± 416	870	1754 ± 273	124
NCI/ADR-RES + Verapamil	45 ± 15	11	183 ± 17	13
SK-OV-3	5.4 ± 0.5	-	11 ± 2	-
SK-OV-3-MDR-1-6/6	486 ± 193	90	127 ± 8	11
SK-OV-3-MDR-1-6/6 + Verapamil	4.4 ± 0.4	0.8	5.3 ± 0.3	0.5

**Table 2 molecules-26-04077-t002:** TGI (nM) of paclitaxel and taccalonolide AF in *MDR*1-expressing ovarian cancer cell lines as compared to isogenic parental lines or in the presence of the Pgp-inhibitor verapamil. Mean ± SEM n = 2–4. Relative resistance calculated as the TGI of the *MDR1-*expressing cell line divided by the TGI value in the isogenic parental line.

Cell Line	Paclitaxel (nM)	Relative Resistance	Taccalonolide AF (nM)	Relative Resistance
OVCAR-8	11 ± 1	-	142 ± 110	-
NCI/ADR-RES	4812 ± 924	458	4291 ± 949	30
NCI/ADR-RES + Verapamil	174 ± 11	17	367 ± 32	2.6
SK-OV-3	10 ± 1	-	27 ± 4	-
SK-OV-3-MDR-1-6/6	~1000	98	412 ± 67	15
SK-OV-3-MDR-1-6/6 + Verapamil	17 ± 1	1.6	11.4 ± 0.6	0.4

**Table 3 molecules-26-04077-t003:** GI_50_ (nM) of paclitaxel and taccalonolide AF in a panel of ovarian cancer cell lines. Mean ± SEM n = 3–7.

Cell Line	Paclitaxel (nM)	Taccalonolide AF (nM)	Ovarian Cancer Cell Line Characterization
ES-2	23 ± 11	11.2 ± 0.5	Clear cell [40], likely HGSOC [41,42,43]
OV-90	5 ± 2	41 ± 6	HGSOC [40,43,44]
OVCAR-3	1.7 ± 0.3	10 ± 3	HGSOC [40,44,45,46]
OVCAR-5	10 ± 3	24 ± 3	HGSOC [47,48,49]
Kuramochi	15 ± 3	17 ± 4	HGSOC [46,47,50]
JHOS-4	4.5 ± 0.7	6.4 ± 0.8	HGSOC [43,50]
OVSAHO	13 ± 6	21 ± 4	HGSOC [46,47,50]

## Data Availability

Appendix A is available online.

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
