# Peer review of "Efficacy of a Covalent Microtubule Stabilizer in Taxane-Resistant Ovarian Cancer Models"

_molecules, 2021, doi:10.3390/molecules26134077_

Round 1

Reviewer 1 Report

I have reviewed the manuscript entitled Efficacy of a Covalent Microtubule Stabilizer in Taxane-Resistant Ovarian Cancer Models. In my opinion, the study was properly undertaken, the results are promising and are well presented and discussed.

My suggestions are:

1.- Include units (nM) in Tables 1 and 2 below the names Paclitaxel and Taccalonolide AF.

2.- In Figure 3, what do authors mean by n=3 and n=2 in figure legend? This is not explained in Materials and Methods.

3.- In Figure 5, legend: “(D) Quantification of NCI/ADR-RES cells in the spleens of animals at the time of AF treatment (day 11) or 29 days after treatment with AF or vehicle (day 40)” seems not to correspond to what is shown in the figure.

Reviewer 2 Report

The major problem is that the authors should describe clearly why select those concentrations of the drugs ( paclitaxel and taccalonolide AF) for in vitro and in vivo test. The criterion for the selection is critical , or the results may be not reasonable.

figure 3 and figure 4, should present or display the photograph of staining colony formation and peeled tumors, not only the column chart.

Figure 5D, words for the X axis need to be modified. The figure legend, “NCI/ADR-RES cells in the spleens of animals at the time of AF treatment (day 11) or 29 days after treatment with AF or vehicle (day 40).” need to be verified?

Reviewer 3 Report

In this manuscript the authors clearly demonstrate that the microtubule stabiliser taccalonolide AF, is less susceptible to Pgp-mediated resistance than paclitaxel and has a higher degree of cellular persistence after drug washout most likely due to its covalent binding to tubulin, the basic building block of microtubules. They demonstrate    also that the taccalonolide AF provides also  a potential advantage in taxane-resistant ovarian cancer models both in cellulo and in mice xenograft model.
Using the same drug resistant tutor cell model in mice xenografts, the same research group has previously shown that targeted delivery of taccalonolides to the tumor could be an effective, longlasting approach to treat drug-resistant tumors (cited ref 20).  Here, in this manuscript they extend their results  and most importantly demonstrate an advantage of taccalonolide AF over paclitaxel in a disseminated metastatic ovarian cancer model, notably the spleen. Based on the results the authors  suggest that the taccalonolides are promising agents to be potentially  used in the localized treatment of overian cancer

The manuscript is well written and merits publication at its current form.

Round 2

Reviewer 2 Report

the revised manuscript is acceptable.